# A Novel Variant of Deformed Wing Virus (DWV) from the Invasive Honeybee *Apis florea* (Apidae, Hymenoptera) and Its Ectoparasite *Euvarroa sinhai* (Acarina, Mesostigmata) in Taiwan

**DOI:** 10.3390/insects14020103

**Published:** 2023-01-18

**Authors:** Jin-Xuan Tian, Wen-Shi Tsai, I-Hsin Sung

**Affiliations:** Department of Plant Medicine, National Chiayi University, Chiayi City 600355, Taiwan

**Keywords:** dwarf honeybee, ectoparasitic mite, honeybee virus, deformed wing, quasispecies

## Abstract

**Simple Summary:**

The deformed wing virus (DWV) has been detected in *Apis florea* and its ectoparasite *Euvarroa sinhai* since they invaded Taiwan. Using a reverse transcription-PCR with specific primers enabled us to ascertain that DWV type A was prevalent among *A. florea* samples. Homology analysis and maximum-likelihood phylogenetic tree between different genotypes disclosed a complete polyprotein nucleotide sequence identity of 88% between isolates from *A. florea*, *E. sinhai*, and DWV type A strains from GenBank. The results demonstrated that a novel variant of deformed wing viruses exists in the invasive species *A. florea* and *E. sinhai*.

**Abstract:**

The invasion of *Apis florea* in Taiwan was first recorded in 2017. The deformed wing virus (DWV) has been identified as a common bee virus in apiculture around the world. Ectoparasitic mites are the main DWV vector for horizontal transmission. However, there are few studies about the ectoparasitic mite of *Euvarroa sinhai*, which has been found in *A. florea*. In this study, the prevalence of DWV among four hosts, including *A. florea*, *Apis mellifera*, *E. sinhai*, and *Varroa destructor*, was determined. The results showed that a high DWV-A prevalence rate in A. florea, ranging from 69.2% to 94.4%, was detected. Additionally, the genome of DWV isolates was sequenced and subjected to phylogenetic analysis based on the complete polyprotein sequence. Furthermore, isolates from *A. florea* and *E. sinhai* both formed a monophyletic group for the DWV-A lineage, and the sequence identity was 88% between the isolates and DWV-A reference strains. As noted above, two isolates could be the novel DWV strain. It cannot be excluded that novel DWV strains could pose an indirect threat to sympatric species, such as *A. mellifera* and *Apis cerana*.

## 1. Introduction

The deformed wing virus (DWV) has been studied extensively, as it is thought to cause colony losses in the Western honeybee *Apis mellifera*, thereby reducing the commercial values of honeybee pollination and products [1]. It is a single-stranded positive-stranded RNA of the *Iflavirus* virus in the family of Iflaviridae, which can prolong its life cycle through mass replication via honeybee pupae and adults or the ectoparasitic mite *Varroa destructor* [2]. Unsuccessful honeybee adults with wing deformities and exposed tongues have been found in front of hive comb cells [3], and even in eclosion survivals, the atrophy of the hypopharyngeal and mandible glands have been observed, which affects their abilities to forage and secrete royal jelly [4].

The DWV has the characteristics of rapid replication and high mutation rate and often exists in the host and in an environment with multiple genotypes or so-called quasispecies [5]. The genotype of the DWV-A lineage includes the *Kakugo virus* (KV) and the so-called *deformed wing virus* (DWV); it has been recognized that it is highly adaptive to various hosts [6], whereas DWV causes wing deformities in honeybees pathologically [7]. DWV-B was isolated from *V. destructor* [8], as it was the main variant of DWV and was originally called *Varroa destructor virus-1* (VDV-1). Subsequent studies have shown at least four variants to be detected in *Varroa* mites [9,10]. The relevant studies also found that various bees, wasps, ants, and other insects incorporated with their symbionts were infected with various DWV genotypes. For example, DWV-A and DWV-C were found in the small hive beetle *Aethina tumida* and stingless bee *Melipona subnitida* [11,12,13].

The DWV genome length is about 10,140 nucleotides (nt) with a poly (A) tail, and 30 nm particle geometry is in icosahedral symmetry [8,14]. The genome structure is monopartite monocistronic with a single open reading frame (ORF) coding a single polyprotein and divided into two parts [15,16]. The major structural protein includes the capsid protein VP1–4 and the non-structural proteins RNA helicase, chymotrypsin-like 3C protease (3C-pro), and an RNA-dependent RNA polymerase (RdRp) which are mapped at the C-terminal region [14,15,16]. A previous study suggested that there were 95% shared amino acids and 84% shared nucleotide identities between DWV-A and DWV-B, and 89.1% and 79% between DWV-A and DWV-C; in addition, the phylogenetic tree showed that DWV-C was independent far from the main branch [17]. To date, though the fourth genotype of DWV-D has been classified, its relevant sequence has not been compared well with the known genotypes [18].

The dwarf honeybee *A. florea* is an invasive alien species (IAS) (Appendix A) that has been brought to attention by its remarkable aggressive expansion via import and export trade worldwide [18,19,20]. The infestation and population density of ectoparasite *Euvarroa sinhai* (Appendix A) in *A. florea* has been studied in terms of geographical origin, whereas little is known about its invasive populations [21,22,23,24]. A study has indicated that *A. florea* can also be infected with DWV [25]. To date, there have been no further reports involving the DWV genotype studies about *A. florea* and its ectoparasite *E. sinhai* in origin or invasive populations. Studies have shown that *E. sinhai* and *V. destructor* in honeybee colonies share life history traits [24,26]. These mites feed on the hemolymph of honeybee host pupae and undermine their immunity and health [27,28]. In addition, they act as a biological vector for the horizontal and vertical transmission of DWV, increasing the viral levels and reducing in-virus diversity [29,30,31,32,33]. Since 2017, *A. florea* has invaded Kaohsiung City, southern Taiwan [20], where many domesticated *A. mellifera* has been reared [32]. In the case of honeybees sharing food resources and interacting with each other in a sympatric area, pathogens carried by one may cause cross-infection through contact behaviors such as mite transfer or competition for nectars [29,30,31]. Thus, the purpose of this study was to detect the prevalence of DWV viruses in invasive *A. florea* and establish whole-genome sequences to compare the differences of viral strains among *A. florea*, *A. mellifera*, *E. sinhai,* and *V. destructor*.

## 2. Materials and Methods

The hot spots of the invasive *A. florea* located in the Fengshan, Gangshan, Gushan, Lingya, Qianzhen, Qijin, and Xiaogang districts in Kaohsiung city were reported by Hsu et al. [20]. From May 2020 to February 2022, 13 nests of *A. florea* were collected periodically in these areas, and between four and eight adult workers in each nest were sampled for monitoring DWV-A and DWV-B. In addition, sympatric *A. florea* and *A. mellifera* foragers were collected by sweeping nets periodically in the open spaces of these areas (Appendix A). A total of 72 and 78 adults of *A. florea* were sampled from nests and open spaces, and 28 of *A. mellifera* were from open spaces, respectively. No symptoms of the deformed wing were found in all the collected honeybee samples. For ectoparasitic mites, five fresh mites of *E. sinhai* were collected from drone cells of the above-mentioned *A. florea* nests, and seven mites of *V. destructor* from *A. mellifera* nests in the apiary of NCYU. Collection details for the sample are given in Appendix A. All samples were deposited in a sample RNA protector (RNAlter) soon after collection [34] and then preserved in a −20 °C freezer until they were isolated.

The total RNA was isolated from single homogenized honeybee midgut or mite bodies’ tissue using RNAzol RT (Molecular Research Center Inc., Cincinnati, OH, USA) [33]. Summarily, mid-gut tissue was homogenized in a 1.5 mL microfuge tube with a disposable tissue grinding pestle in 500 μL RNAzol RT and 200 μL of added deionized water (diH2O); the resulting mixture was shaken vigorously and allowed to stand at room temperature for 10 min before the sample mixture was centrifuged at 12,000 rpm for 15 min. The supernatant liquid (500 μL) was then placed in a microfuge tube, and an equal volume of isopropanol (500 μL) was added, and after mixing, the sample solution was placed at room temperature for 15 min. The mixed solution was then centrifuged (12,000 rpm for 10 min), and the pellet was washed twice with 75% ethanol and then repelleted (4000 rpm for 3 min). The supernatant was removed, and the dried pellet was reconstituted in deionized water (diH2O) 50 μL.

Sample RNA was used to synthesize cDNA with an oligo (dT) 18 primer (Protech, Taipei, Taiwan) by a HiScript I TM First Strand cDNA Synthesis Kit (BIONOVAS Biotechnology, Toronto, Ontario, Canada). PCR reactions were performed in a 25 µL mixed solution using a 2× red PCR Master mix kit (Ampliqon). For DWV-A and DWV-B, the three primers used to sequence the RdRp-encoding region were DWV-F (GGATGTTATCTCTTGCGTGGA), DWV-R (CGATAATAATTTCGAACGCTGA) (412 bp) [14]; KV-F (GGACTGAACCAAATCCGATGTCATCACG), and KV-R (TCTCAAGTTCGGGACGCATTC) (378 bp) [7] and VDV-1-F (TGGCTAATCGACGTAAAGCA) and VDV-R (ACTAATCTCTGAGCCAACACGT) (195 bp) [35]. We chose a DWV, KV, and VDV-1 positive *A. mellifera* sample from the apiary of NCYU as a positive control.

Ten sets of primers were designed to amplify overlapping PCR products, which comprised the complete polyprotein genome sequence of targeted DWV isolates. The sequences, orientations, and locations of the primers, as well as the expected product sizes, are shown in Table 1. The thermal cycling program followed: one cycle at 94 °C for 5 min, 35 cycles at 94 °C for 30 s, 50 °C for 30 s, and 72 °C for 30 s, and an elongation cycle at 72 °C for 5 min. The reaction was then held at 4 °C. The PCR-amplified product was electrophoresed in a 2% agarose gel containing 0.5 μg/mL of ethidium bromide and was visualized under UV light. 

PCR reactions used a Q5 High-Fidelity DNA Polymerase Kit (New England Biolabs, Ipswich, MA, USA) and 10 mM dNTPs. Each reaction contained a 5x Q5 buffer, 2.5 mM dNTPs, Q5 high-fidelity DNA polymerase, forward and reverse DWV-variant-specific primer (Table 1), diH2O, and cDNA template. The thermocycler was set to initial activation at 98 °C for 30 s, followed by 30 cycles of denaturation at 98 °C for 10 s, annealing at 52 °C for 30 s, and extension at 72 °C for 3 min, followed by a final extension at 72 °C for 5 min. The purified gene fragments were separately cloned with a pGEM-T easy vector (Promega) kit and transformed into competent cells, *Escherichia coli* (DH5α, Smobio Champion *E. coli* Transformation Kit). The transformants (*E. coli*) were selected by blue-white selection on the plates containing 5% X-gal (20 mg/mL), Ampicillin (100 μ*g*/mL), and IPTG (50 Mm)) and were screened as positive clones by colony PCR. The plasmid was isolated and then restricted the enzyme digestion of plasmid DNAs with EcoR1. For each virus, 3–5 clones were picked up for Sanger sequencing. The cDNA clones were reamplified from the respective plasmid with a pair of T7P and SP6 by automated DNA sequencer at Genomics BioSci & Tech Co., Ltd. Company (New Taipei City, Taiwan). All the sequences were submitted to GenBank (accession nos. OP889266–OP889269).

Based on the complete polyproteins to construct a homology tree and phylogenetic tree, the phylogenetic relationships among the strains of the honeybee viruses and other close groups available in GenBank were initially assessed using BLAST [36]. The identification of all the gene domains referred to the predicted protease sites for the DWV reference sequence (AJ489744.2) [14]. The nucleotide (nt) and amino acid (aa) sequences of the viruses were aligned with sequence analyses and phylogenetic analyses by using DNAMAN and MEGA X [37,38]. The maximum-likelihood phylogenetic tree (model TN93 + G; 1000 replicates) was also constructed from sequences of DWV-A, DWV-B, DWV-C, and sequences of virus isolates in this study (Table 2).

## 3. Results

### 3.1. Prevalence of DWV-A and DWV-B in Honeybees and Mites

All samples containing DWV were amplified by the PCR using the primers KV-F/-R and DWV-F/-R, and it was confirmed by gel electrophoresis that the KV primer showed higher sensitivity than DWV in *A. florea*, *A. mellifera,* and *E. sinhai* (Figure 1; Table 3). For DWV-A testing, *A. florea* showed partial/all positives in the samples collected from nests and open spaces in different locations, whereas only five locations were partial/all positive in *A. mellifera* (Table 3). Meanwhile, all the samples were negative in DWV-B testing. A total of 68 (94.4%) *A. florea* nest samples were positive for KV, and 54 (69.2%) were in open-space samples (Table 4). Though *A. mellifera* showed a lower portion of positive results (32.1%), all samples from *E. sinhai* and *V. destructor* were positive for KV.

### 3.2. Genome Sequence of DWV Variant in A. florea and E. sinhai

The DWV homology tree and phylogenetic tree were constructed from complete poly protein nt sequences of *A. florea* (AF01T243 (AF01), accession nos. OP889266), *A. mellifera* (AM01TGS8 (AM01), accession nos. OP889268), *E. sinhai* (ES01TFS11 (ES01), accession nos. OP889267), and *V. destructor* (VD01Tvd1 (VD01), accession nos. OP889269) in this study, along with the previously reported DWV isolates (Table 2) and outgroup Black queen cell virus (BQCV) (MW390818.1) from the GenBank [39]. The DWV isolates in the homology tree were closely related to form a monophyletic subclade, with a relatively large branch of 25 isolates in lineage I and an isolate of DWV-C in II (Figure 2A). Lineage I divided into two subgroups for *A* and *B* AF01, ES01, AM01, VD01, and 17 DWV-A isolates in subgroup *A*, whereas four DWV-B isolates were placed in group *B*. The two master lineages showed 80% sequence homology, while two subgroups from lineage I showed 85% sequence homology. Within subgroup *A*, the isolates were further subdivided into two branches: one for AM01, VD01, and 17 DWV-A isolates and the second for AF01 and ES01 isolates. Moreover, the two branches shared 88% sequence identity. With respect to the maximum-likelihood phylogenetic tree (Figure 2B), the 26 DWV isolates (including AF01, ES01, AM01, VD01) were divided into two branches for lineages i and ii. Lineage i contained two subgroups for *a* and *b* AM01, VD01, and other 17 DWV-A isolates in subgroup *a*, compared to four DWV-B isolates and one DWV-C isolate in *b*. The AF01 and ES01 isolates were found in lineage ii. Furthermore, the phylogenetic tree that was constructed for RdRp nt sequences of *A. florea* (AF01), *A. mellifera* (AM01), *E. sinhai* (ES01), and *V. destructor* (VD01) in this study, along with the previously reported DWV isolates (Table 2) and DWV-C reference strains as the outgroup sequence, formed the result that AF01 and ES01 isolate formed a subcluster distinct from the clusters of the DWV-A (Figure 3).

Based on the present results, it can be seen that DWV isolates from *A. florea* and its ectoparasite *E. sinhai* formed another distinct clade. Genome regions of alignment need to be studied for further analysis: (1) The ORF is predicted to encode a 2896 aa polyprotein. (2) 950 aa capsid proteins; (3) 713 aa 3C + RdRp; (4) 472 aa helicase; (5) 211 aa leader protein; (6) 320 nt 3′ UTR (without poly-A); and (7) 1135 nt 5′ UTR. Homology analyses performed by nucleotides and amino acids of 3′ UTR, 5′ UTR, the polyprotein, VP1-4 of the capsid protein (structural protein), the leader protein of structural polyprotein, helicase, 3-chymotrypsin-like protease, and the 3-chymotrypsin-like protease (3C + RdRp) region were referred to the complete DWV-A genome sequence (AJ489744.2). Here, we only describe the nucleotide identity in each region of the DWV genome. The findings of the complete nucleotide (nt) and amino acid (aa) sequence identities are in alignment with AF01, and the corresponding regions of the four isolates (ES01, DWV-A (AJ489744.2), DWV-B (AY251269.2) and DWV-C (CEND01000001.1)) was shown (Table 5). For the 3′ UTR and 5′ UTR regions, there was a 99.7–99.9% shared nt identity within the two isolates of AF01 and ES01, which shared 94–95.6% nt similarity with DWV-A, 82.6–89.9% with DWV-B, and 82.8–86.9% with DWV-C. For the polyprotein region, there was 98.2% shared nt identity within AF01 and ES01 (complete sequence in Appendix A), while the two isolates shared 89% with DWV-A, 83% with DWV-B, and 79.1% with DWV-C. For the leader protein region, there was a 99.5% shared nt identity within AF01 and ES01, while two isolates shared an 82% similarity with DWV-A, 74.8% with DWV-B, and 65.2% with DWV-C. For the capsid protein region, there was a 99.8% shared nt identity within AF01 and ES01, while two isolates shared 86.9% with DWV-A, 82.6% with DWV-B, and 79.3% with DWV-C. For the helicase region, there was a 93.5% shared nt identity within AF01 and ES01 (Appendix A), while two isolates shared 94.1% with DWV-A, 87.5% with DWV-B, and 82.1% with DWV-C. It is also worth noting that the AF01 helicase sequence was significantly lower than the other regions. This result was highlighted in the disparity between AF01 and ES01. For the 3C + RdRp region, there was a 99.9% shared nt identity within AF01 and ES01, while the two isolates shared an 89.2% similarity with DWV-A, 84.5% with DWV-B, and 82% with DWV-C.

## 4. Discussion

In this study, we emphasized only the prevalence and variants of DWV in the invasive honeybee *Apis florea* and its ectoparasite *Euvarroa sinhai* in Taiwan. Four master variants have been described in the DWV of the quasispecies [7,14,17,40]. Unfortunately, the complete viral polyprotein sequence from *A. florea* and *E. sinhai* has not been reported to the GenBank database. According to the International Committee on Taxonomy of Viruses (ICTV), species of the *Iflaviridae* are demarcated by amino acid identity in the sequence of the capsid proteins (CPs) when the strains of a species are above 90% [41]. In light of this, an over 90% sequence homology was displayed between the isolates from *A. florea* (AF01T243), *E. sinhai* (ES01TFS11), and the DWV-A strain (AJ489744.2) with regard to the CPs; therefore, we conclude that two virus isolates exist in the new variant of DWV. In addition, the phylogenetic comparison of DWV from various species in this study suggests that there are at least three distinct main groups within the DWV quasispecies. DWV isolates found in Taiwan were clustered within the DWV-A clade. Each one of the isolates from *A. florea* and *E. sinhai* in this study formed a monophyletic subgroup with other strains of the DWV-A, and the similarity of the two isolates was only 89% with DWV-A sequences. The 3C-pro and RdRp regions of the DWV genome from *A. florea* and *E. sinhai* tend to be highly conserved among the RNA viruses [42]. Therefore, they have usually been used as a reliable protein for the construction of phylogenetic trees and for the classification of subtypes of RNA viruses [42,43,44]. In the present study, the phylogenetic analysis of 3C + RdRp showed that the two novel isolates all harbored a distinct DWV-A-clade and formed a distinct subgroup. They might share a common ancestor and could have evolved independently in secluded geographic regions.

Turning to the experimental result on the sequence homology between the isolate from *A. florea* and *E. sinhai*, the sequence similarity in the helicase coding regions was the lowest (93.5%). According to previous studies, helicase is the most highly conserved region and key recombinant hotspot of DWV genomes, but whether there is a recombination between isolates in the helicase region needs to be further explored [14,35]. Specifically, we observed an additional difference in the helicase region between isolates from *A. florea* and *E. sinhai*. As a result of all the above results combined, it was found that there was still genetic variation between the two isolates and DWV-A isolates. We have evidence that the isolates from *A. florea* and *E. sinhai* are both new variants of DWV. In fact, at least a slight sequence difference in the genome, or the substitution of a few amino acids, is enough to produce different toxicities in certain types of viruses [45]. In the future, artificial infection experiments using infectious clones of variants will be needed to verify these possibilities. In this study, a high prevalence of DWV was observed in *A. florea* without apparent symptoms and in *E. sinhai* samples. It can also be observed that the nucleotide and amino acid sequences between the *A. florea* and *E. sinhai* DWV isolates were highly identical. From the above result, we can determine the pathogenic correlation between mites and honeybees, but there is not yet strong enough evidence of which is the original virus carrier. Previous studies have indicated that *A. florea* and *A. mellifera* have been found to prey on honey sources from hives, and on top of that, *A. florea* carries the ectoparasitic mite *E. sinhai* and has also been shown to survive on *A. mellifera* and *A. cerana* [22,46]. In Thailand, *E. sinhai* has been found in the nests of *A. mellifera* [46]. The results of this study found a novel DWV variant from samples of *A. florea* and *E. sinhai*. The *V. destructor*, which was originally transferred from *A. cerana* to *A. mellifera*, contributed to a significant increase in the prevalence of DWV in the honeybee colony, a decrease in the diversity of the deformed wing virus, and the dominance of a single DWV genotype [47].

The presence of different species of bees and their associated parasitic mites in sympatric areas may facilitate the exchange of parasites between them and the simultaneous infestation of multiple mites at the population or individual level [48,49]. The dwarf honeybee *A. florea* and its acquired mite *E. sinhai* are invasive species in Taiwan and several other countries [50,51], and it is also worth noting that they carry new DWV variants. However, few studies have evaluated the influences of *Euvarroa* mites on honeybees. The life cycle and feeding behavior of *E. sinhai* are ecologically similar to the *V. destructor* [23]. As far as *E. sinhai* is concerned, it may be a dominant pathogen transmitter and lead to a chain reaction of cross-infection in the overlapping niches of sympatric honeybee species, which indirectly affects the evolution of the virus [52,53,54], such as the mite *Tropilaelaps mercedesae* and *V. destructor* [55,56].To sum up, based on the comparison results of nucleotide and amino acid sequences in each coding gene region and the nucleotide sequences in the non-translated region, it can first be concluded that the virus has high genetic diversity traits when targeting different hosts in *A. mellifera* and *A. florea* [57], and secondly, the phenomenon of high homology between the virus isolates of honeybees and their ectoparasitic mites was found.

## 5. Conclusions

Our data indicate that the new virus variants are suspected to be carried by invasive *A. florea* and its ectoparasitic mite *E. sinhai*. Moreover, mites might be a biological vector and open up new avenues of inter-taxa virus transmission. Future research should consider the potential effects of new virus strains more carefully, and it should be investigated whether it has become a new variant of the deformed wing virus so as to establish the invasion risk level of the invasive *A. florea* and avoid endangering the apiculture of Taiwan and the world.

## Figures and Tables

**Figure 1 insects-14-00103-f001:**
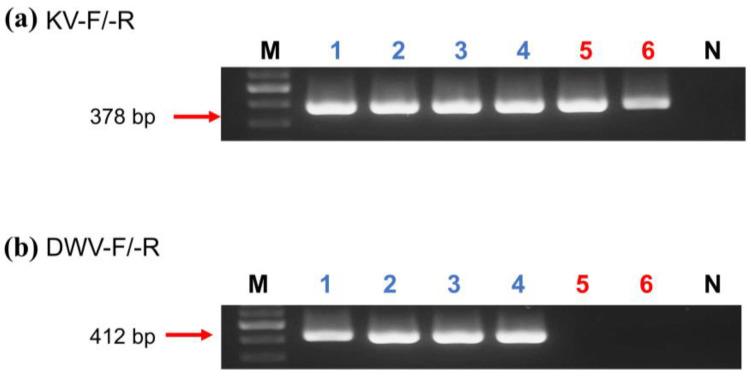
Gel electrophoresis of KV ((**a**), 378 bp) and DWV ((**b**), 412 bp) products amplified by selective primers KV-F/-R and DWV-F/-R, respectively. (**a**,**b**) Lane M: 100 bp Nautia DNA Ladder (Nautia); Lane N: negative control; Lanes 1–3: *A. mellifera*; Lanes 4: *V. destructor*; Lanes 5: *A. florea*; and Lanes 6: *E. sinhai*.

**Figure 2 insects-14-00103-f002:**
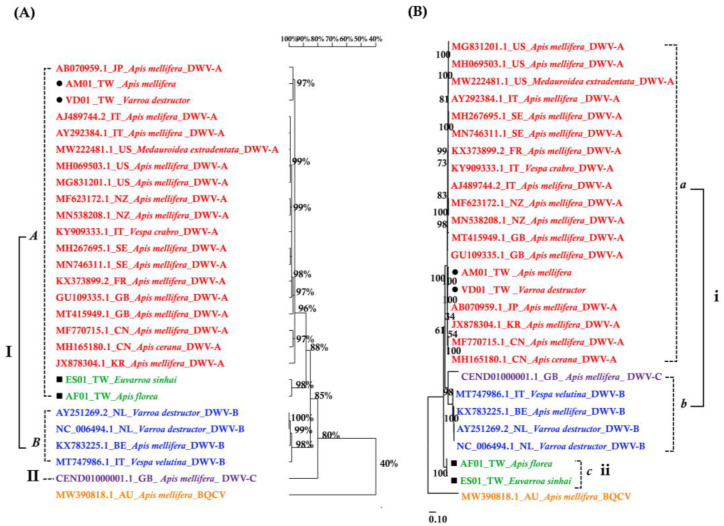
(**A**) Homology tree constructed based on the complete polyprotein sequence of four samples in this study; the 22 DWV-associated strains and Black queen cell virus complete genome (MW390818.1) were from the NCBI database. (**B**) Maximum-likelihood phylogenetic tree (Tamura-Nei model; 1000 replicates) inferred from complete polyprotein sequence.

**Figure 3 insects-14-00103-f003:**
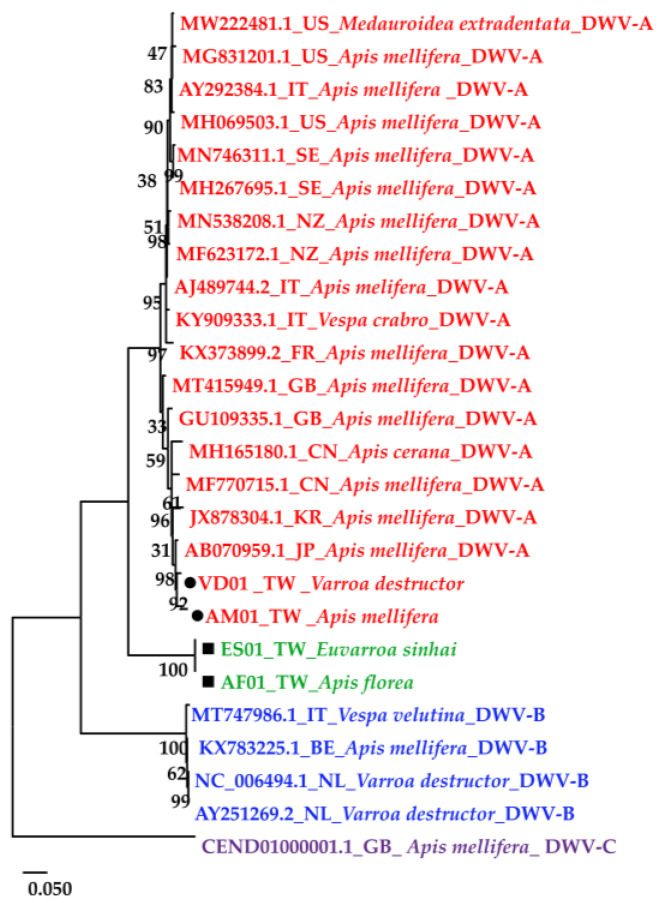
Maximum-likelihood phylogenetic tree (Tamura-Nei model; 1000 replicates) constructed based on the RdRp region sequence of four isolates in this study; the 22 DWV-associated strains and complete genome were from the NCBI database.

**Table 1 insects-14-00103-t001:** The primers designed for overlapping sequences to ensure the complete polyprotein sequencing of the selected DWV genotypes.

Primer Pairs	Sequence (5′-3′)	Amplicon Size (nt)	Nucleotide Position *
DWV-F4 DWV-R4	CATTGGTATGCTCCGTTGACTG CTCTTGCGCCATGGTCCAC	2330	6866–9199
DWV-C4-1	CCTGGTAGTAAGTGGCG		
DWV-F7 DWV-R7	GCGATTTATGCCTTCCATAGCG CTCWGGYTTTGCCTGCACCG	1780	1–1777
DWV-F8 DWV-R8	CATATAGACCATGGTGGGTGCG CGCATCTTTGCTGCCTGAGC	1550	3144–4698
DWV-F9 DWV-R9	GCGCTGCATCTAGTTATGC GCACGATAGGAGAATGGACC	1380	4416–5795
DWV-F9-2 DWV-R9-2	GTGATGCTGTGTCTACTGG GGAGTACGACTCGCACG	1340	4270–5637
DWV-F12 DWV-R12	CCAGGACCTGATGGCGAG GCTATGCCACACTCCAGC	1640	1493–3433
DWV-F12-2 DWV-R12-2	AKCTAATCCGGTGCAGGC GTACTAGGAGCATCAGTCG	1460	1747–3211
DWV-F13 DWV-R13	TATCTTGGAATACTAGTGCTGG TATCTTGGAATACTAGTGCTGG	1470	8614–10,079
DWV-F7-3 DWV-R11-3	CTACGGTACGTTACGTTCG GGACCAGTAGCACTCATC	1560	875–2433
Oligo(dT)18	d (TTT TTT TTT TTT TTT TTT)		

* Nucleotide positions refer to the published DWV-A sequence with GenBank accession number AJ489744.2.

**Table 2 insects-14-00103-t002:** DWV sequences published in GenBank used to construct phylogenetic trees.

Accession Number	Host Species	Geographic Origin
AJ489744.2	*Apis mellifera*	Italy (IT)
AY292384.1	*Apis mellifera*	Italy (IT)
MT415949.1	*Apis mellifera*	United Kingdom (GB)
GU109335.1	*Apis mellifera*	United Kingdom (GB)
MG831201.1	*Apis mellifera*	United States (US)
MH069503.1	*Apis mellifera*	United States (US)
MF623172.1	*Apis mellifera*	New Zealand (NZ)
MN538208.1	*Apis mellifera*	New Zealand (NZ)
MH267695.1	*Apis mellifera*	Sweden (SE)
MN746311.1	*Apis mellifera*	Sweden (SE)
KX373899.2	*Apis mellifera*	France (FR)
AB070959.1	*Apis mellifera*	Japan (JP)
JX878304.1	*Apis mellifera*	South Korea (KR)
MF770715.1	*Apis mellifera*	China (CN)
KX783225.1	*Apis mellifera*	Belgium (BE)
CEND01000001.1	*Apis mellifera*	United Kingdom (GB)
MH165180.1	*Apis cerana*	China (CN)
KY909333.1	*Vespa crabro*	Italy (IT)
MT747986.1	*Vespa velutina*	Italy (IT)
AY251269.2	*Varroa destructor*	Netherlands (NL)
NC_006494.1	*Varroa destructor*	Netherlands (NL)
MW222481.1	*Medauroidea extradentata*	United States (US)

**Table 3 insects-14-00103-t003:** Detection of DWV-A and DWV-B in workers of *Apis florea* and *A. mellifera* by RT-PCR.

Sample Source	Sample Size (Af/Am)	Location	*A. florea* (Af) *			*A. mellifera* (Am) *		
			DWV-A		DWV-B	DWV-A		DWV-B
			DWV	KV		DWV	KV	
**OPEN SPACE**								
	5/3	Qianzhen	negative	positive	negative	negative	negative	negative
	5/3	Qianzhen	positive	positive	negative	negative	negative	negative
	6/2	Qianzhen	negative	positive	negative	negative	negative	negative
	5/0	Qianzhen	negative	positive	negative			
	5/2	Gushan	negative	positive	negative	negative	positive	negative
	5/0	Gushan	negative	positive	negative			
	4/4	Fengshan	negative	positive	negative	negative	positive	negative
	5/4	Fengshan	negative	positive	negative	negative	positive	negative
	8/0	Fengshan	negative	positive	negative			
	8/1	Qijin	negative	positive	negative	negative	negative	negative
	14/2	Qijin	negative	positive	negative	positive	negative	negative
	4/4	Gangshan	negative	positive	negative	negative	positive	negative
	4/3	Lingya	negative	positive	negative	negative	negative	negative
**NEST**								
	4/0	Xiaogang	positive	positive	negative			
	6/0	Xiaogang	negative	positive	negative			
	4/0	Qianzhen	positive	positive	negative			
	4/0	Qianzhen	positive	positive	negative			
	4/0	Qianzhen	positive	positive	negative			
	5/0	Gushan	negative	positive	negative			
	7/0	Gushan	negative	positive	negative			
	6/0	Lingya	negative	positive	negative			
	6/0	Lingya	negative	positive	negative			
	8/0	Lingya	positive	positive	negative			
	8/0	Fengshan	negative	positive	negative			
	6/0	Qijin	negative	positive	negative			

* positive: DWV-A or DWV-B detected by PCR in worker samples; negative: DWV-A or DWV-B not detected by PCR in worker samples; primer use: DWV-F/-R [14], KV-F/-R [7].

**Table 4 insects-14-00103-t004:** Prevalence of DWV-A in *Apis florea*, *Euvarroa sinhai*, *A. mellifera* and *Varroa destructor*.

Sample Source	Species	Sample Size	DWV-A (DWV/KV) * (%)	
			DWV Positive	KV Positive
Nest	*A. florea*	72	11 (15.3%)	68 (94.4%)
Open Space	*A. florea*	78	1 (1.3%)	54 (69.2%)
Open Space	*A. mellifera*	28	1 (3.6%)	9 (32.1%)
Nest	*E. sinhai*	5	0	5 (100%)
Nest	*V. destructor*	7	0	7 (100%)

* primer use: DWV-F/-R [14], KV-F/-R [7].

**Table 5 insects-14-00103-t005:** Percentage nucleotide and amino acid sequence’s identity in an alignment between 243 isolate from *Apis florea*, three genotypes, and other isolates in this study.

AF01 (AF01T243)	Sequence Identity (%)			
	ES01 (ES01TFS11)	DWV-A (AJ489744.2)	DWV-B (AY251269.2)	DWV-C (CEND01000001.1)
Nucleotide sequence				
3′ UTR	99.9	95.6	89.9	86.9
5′ UTR	99.7	94.0	82.6	82.8
Entire polyprotein	98.2	89.0	83.0	79.1
Leader protein	99.5	82.0	74.8	65.2
Capsid protein	99.8	86.9	82.6	79.3
Helicase	93.5	94.1	87.5	82.1
3C + RdRp	99.9	89.2	84.5	82.0
Amino acid sequence				
Entire polyprotein	99.4	95.2	93.6	88.6
Leader protein	99.5	84.4	78.7	66.8
Capsid protein	99.9	96.0	95.4	89.7
Helicase	98.9	98.1	97.5	93.9
3C + RdRp	100	97.6	96.8	93.8

## Data Availability

The viral polyprotein sequence of DWV isolates in this study were submitted to GenBank with an accession number (OP889266-OP889269).

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
