# Peer review of "A Novel Variant of Deformed Wing Virus (DWV) from the Invasive Honeybee Apis florea (Apidae, Hymenoptera) and Its Ectoparasite Euvarroa sinhai (Acarina, Mesostigmata) in Taiwan"

_insects, 2023, doi:10.3390/insects14020103_

Round 1

Reviewer 1 Report

The manuscipt described a novel strain of DWV from Apis florea and its parasite, which give new information about this prevalence virus. The results are quite straight forward. There are some minor concerns need to be elucidated as following:

Line21: There should be a comma between  "Apis mellifera" and  "E. sinhai".

-Fig.1 "Lane B: negative control", As "M" stands for "Molecular marker", I suggest that the "N", the initial letter of "Negative", is better than "B" in the figure. KV and DWV could be print on the side of gel images for readers' convenience.

L116-117: For the whole genome sequencing , authors used 10 sets of primers, which inculde the "DWV-F7/R7" for the position of 1-1777 in the 5' end, and DWV-F13/R13 for 8614-10079. As in Line 200-204, these two primers can amplify the both end of the genome, Generally 3'-and 5'-Race technique are preferred for the both end amplification, Could authors explain why the RACEs  is not necessary?

L115: The primers VDV-1-F/R are used for RdRP detection in L115, though there was not information about its sensitivity comparing to the other two in paragraph "3.1", furthermore, if KV-F/R is better, whether the results in table 3 and 4 were done by KV primers or others, this should be noted in the table. 

Table 3: The second line  should be a broken in the middle between "A. florea (Af)*" and "A. mellifera (Am)*" to seperate two parts.

Line 274: "..., high detection of DWV..."  changes to "...,high prevalence of ..."

Author Response

  1. author responses

We thank the reviewer for their appraisal of the work, and are pleased that the description of the result is well understood.

  1. Comments and Suggestions for Authors

We thank the reviewer for their time and are glad they think the result is useful and relevant.

Based on the reviewer’s comments, we revised the results and explained as follows. We also use tracking revisions in MS so that reviewers can clearly understand the revisions we made based on the comments.

  • Line21: There should be a comma between "Apis mellifera" and  " sinhai".

RE: The content of the study has been modified. Please see revisions in our manuscript.

  • -Fig.1 "Lane B: negative control", As "M" stands for "Molecular marker", I suggest that the "N", the initial letter of "Negative", is better than "B" in the figure. KV and DWV could be print on the side of gel images for readers' convenience.

RE: The figure 1 of the study has been modified. Please see revisions in our manuscript.

  • L116-117: For the whole genome sequencing, authors used 10 sets of primers, which inculde the "DWV-F7/R7" for the position of 1-1777 in the 5' end, and DWV-F13/R13 for 8614-10079. As in Line 200-204, these two primers can amplify the both end of the genome, Generally 3'-and 5'-Race technique are preferred for the both end amplification, Could authors explain why the RACEs is not necessary?

RE: In this study, we focus on the structural polyprotein sequence of DWV, there is no limit to define complete 5'-sequences. However, we still presented and analyzed the sequence homology of partial 5' end we obtained.

  • L115: The primers VDV-1-F/R are used for RdRP detection in L115, though there was not information about its sensitivity comparing to the other two in paragraph "3.1", furthermore, if KV-F/R is better, whether the results in table 3 and 4 were done by KV primers or others, this should be noted in the table.

RE: Because of the low detection rate of VDV-1 (DWV-B) and high similarity of DWV-A (DWV, KV) sequences, this experiment was required to confirm the efficiency of the primer for DWV and KV.

  • Table 3: The second line should be a broken in the middle between "A. florea (Af)*" and "A. mellifera (Am)*" to seperate two parts.

RE: The table of the study has been modified. Please see revisions in our manuscript.

Line 274: "..., high detection of DWV..."  changes to "...,high prevalence of ..."

RE: The content of the study has been modified. Please see revisions in our manuscript.

Reviewer 2 Report

Overall, an interesting paper examining the classification of novel DWV isolates in A. florea and E. sinhai. I suggest the following comments be addressed prior to the paper being published:

The manuscript would benefit from editing by a native English speaker. In places, the meaning of sentences is confusing, sometimes incomprehensible. With editing, I’ve no doubt this paper could be well placed for this journal.

The methods section switches from past to present tense throughout. Edit to a single tense to improve flow.

What positive control was used in PCRs (e.g. plasmid construct/ G-Blocks)? Include improving the reliability of results, most notably where no DWV-B was detected in any samples.

Was replicating strand of these novel isolates detected in either E. sinhai, or A. florea? Looking at this (via PCR or otherwise) may help indicate where the virus is replicating and give more weight to the findings.

Include images of whole gels, even in supplemental, along with the sequence of the novel isolates. Whilst available on NCBI, the manuscript may benefit from the inclusion of the sequences.

The phylogenetic trees are in fact rooted despite the authors stating otherwise. For instance, BQCV is the root in figure 5B. It’s likely that MEGA defaulted the root as this, but regardless, the tree is rooted.

Whilst the authors correctly identified that the viral genome in question contains certain mutations, indicating variants, no proof was given of the difference of isolates from DWV-A in phenotypic properties such as antigenicity, transmissibility, or virulence. As such, terming this a new strain may not be appropriate without further research/ justification in the manuscript. “Novel variant” may be more appropriate.

The authors provided a concise summary of the potential impacts of the novel isolates. In some places, the discussion appears a little repetitive and could benefit from expansion to provide more insight into the data.

Author Response

  1. author responses

We thank the reviewer for their appraisal of the work, and are pleased that the description of the result is well understood.

  1. Comments and Suggestions for Authors

We thank the reviewer for their time and are glad they think the result is useful and relevant.

Based on the reviewer’s comments, we revised the results and explained as follows. We also use tracking revisions in MS so that reviewers can clearly understand the revisions we made based on the comments.

  • The manuscript would benefit from editing by a native English speaker. In places, the meaning of sentences is confusing, sometimes incomprehensible. With editing, I’ve no doubt this paper could be well placed for this journal.

RE: As recommended, the manuscript was edited with the assistance of a native English speaker.

  • The methods section switches from past to present tense throughout. Edit to a single tense to improve flow.

RE: The content of the study has been modified. Please see revisions in our manuscript.

  • What positive control was used in PCRs (e.g. plasmid construct/ G-Blocks)? Include improving the reliability of results, most notably where no DWV-B was detected in any samples.

RE: One sample 197 from apiary of NCYU with a positive PCR for VDV-1 as positive control. The content of the study has been modified. We add explain “We chose a DWV, KV and VDV-1 positive A. mellifera sample from apiary of NCYU as a positive control.” In line 123.

  • Was replicating strand of these novel isolates detected in either sinhai, or A. florea? Looking at this (via PCR or otherwise) may help indicate where the virus is replicating and give more weight to the findings.

RE: The experiment has not yet been carried out.

  • Include images of whole gels, even in supplemental, along with the sequence of the novel isolates. Whilst available on NCBI, the manuscript may benefit from the inclusion of the sequences.

RE: We will attach the relevant sequences (refer to the Table S2).

  • The phylogenetic trees are in fact rooted despite the authors stating otherwise. For instance, BQCV is the root in figure 5B. It’s likely that MEGA defaulted the root as this, but regardless, the tree is rooted.

RE: The content of the study has been modified. Please see revisions in our manuscript.

  • Whilst the authors correctly identified that the viral genome in question contains certain mutations, indicating variants, no proof was given of the difference of isolates from DWV-A in phenotypic properties such as antigenicity, transmissibility, or virulence. As such, terming this a new strain may not be appropriate without further research/ justification in the manuscript. “Novel variant” may be more appropriate.

RE: The content of the study has been modified. Please see revisions in our manuscript.

  • The authors provided a concise summary of the potential impacts of the novel isolates. In some places, the discussion appears a little repetitive and could benefit from expansion to provide more insight into the data.

RE: The content of the study has been modified. Please see revisions in our manuscript.

Text added to discussion:

Discussion [...]  In fact, at least a slight sequence difference of genome, or the substitution of a few amino acids is enough to produce different toxicities in certain types of viruses [45]. In the future, artificial infection experiments using infectious clones of vaiants will be needed to verify these possibilities. [...] The presence of different species of bees and their associated parasitic mites in sympatric area may facilitate the exchange of parasites between them and the simultaneous infestation of multiple mites at the population or individual level [48,49]. Red dwarf honeybee A. florea and its acquired mite E. sinhai from are invasive species in the Taiwan and a few countries [50, 51], and it is also worth noting that they carry new DWV variants from this study. However, few studies have evaluated the influences of Euvarroa mites on honeybees. The life cycle and feeding behavior of E. sinhai is ecologically similar to V. destructor [23]. As far as E. sinhai is concerned, it may be a dominant pathogen transmitter and lead to a chain reaction of cross-infection in overlap niches of sympatric honeybee species, which indirectly affects the evolution of the virus to attention [52–54], such as the mite Tropilaelaps mercedesae and V. destructor.[55,56]